# Fucoidan But Not 2′-Fucosyllactose Inhibits Human Norovirus Replication in Zebrafish Larvae

**DOI:** 10.3390/v13030461

**Published:** 2021-03-11

**Authors:** Malcolm Turk Hsern Tan, Yan Li, Mohamad Eshaghi Gorji, Zhiyuan Gong, Dan Li

**Affiliations:** 1Department of Food Science & Technology, Faculty of Science, National University of Singapore, Singapore 119077, Singapore; malcolmtth@u.nus.edu (M.T.H.T.); mohamad.e.gorji@u.nus.edu (M.E.G.); 2Department of Biological Sciences, Faculty of Science, National University of Singapore, Singapore 119077, Singapore; dbsliya@nus.edu.sg (Y.L.); dbsgzy@nus.edu.sg (Z.G.)

**Keywords:** norovirus, fucoidan, 2′-fucosyllactose, antiviral

## Abstract

Human noroviruses (hNoVs) cause heavy disease burden worldwide and there is no clinically approved vaccination or antiviral hitherto. In this study, with the use of a zebrafish larva in vivo platform, we investigated the anti-hNoV potentials of fucoidan (from brown algae *Fucus vesiculosus*) and 2′-Fucosyllactose (2′-FL). As a result, although both fucoidan and 2′-FL were able to block hNoV GII.4 virus-like particle (VLPs) from binding to type A saliva as expected, only fucoidan, but not 2′-FL, was able to inhibit the replication of hNoV GII.P16-GII.4 in zebrafish larvae, indicating the possible needs of higher molecular weights for fucosylated carbohydrates to exert anti-hNoV effect.

## 1. Introduction

Human noroviruses (hNoVs) are increasingly identified as an important cause of acute gastroenteritis outbreaks associated with great disease burden worldwide. In 2015, the World Health Organization (WHO) estimated that 684 million diarrheal disease cases were caused by hNoVs annually, amongst which 212,000 deaths occurred [1]. Since hNoVs are frequently linked with foodborne outbreaks, it is also regarded as the “number 1” cause of foodborne illnesses [1,2,3]. Similar to many other viruses, hNoV-caused infection is usually self-limiting within a few days for most healthy people, while severe and long-lasting symptoms and even death could be induced in the vulnerable population [4].

Vaccination, despite of significant research efforts dedicated for many years, remains infeasible as hNoVs evolve extremely rapidly [5]. As for antivirals, none has yet been clinically approved, and the majority of candidates are still in the early stages of preclinical development, as reviewed by Netzler et al. (2018) [6]. Thus, there is urgent need to screen for alternative solutions based on solid scientific evidence to mitigate NoV infection. especially for the vulnerable population.

Since histo-blood group antigens (HBGA) have been shown to have close correlations with hNoV infection [7,8], the in vitro HBGA-binding block assay has been used in many studies to screen for anti-hNoV candidates [9,10,11,12]. As hNoV cultivation remains to be challenging, not many of the candidates have been further validated [13]. In 2019, a zebrafish (*Danio rerio*) larva platform was reported to be able to support replication of multiple hNoV strains [14]. This is of great interest for its feasibility of high-throughput and cost-effectiveness. In fact, zebrafish, which share remarkable genetic, physiologic and pharmacologic similarities to humans [15], have been used as a versatile research organism that can be experimentally infected with many viruses [16,17,18]. It has been proposed to be used for the discovery of new viruses recently [19].

In this study, we applied the zebrafish larva platform to evaluate the anti-hNoV effect of fucoidan and 2′-Fucosyllactose (2′-FL). Both carbohydrates have been reported as putative anti-hNoV candidates due to their fucose contents and thus specific hNoV binding ability [20,21,22,23,24]. Specifically, 2′-FL was observed to bind to NoV GII.4 and GII.10 capsid at the same pockets with HBGA [23]. Fucoidan was proposed as a more efficient anti-NoV candidate due to its fucose multivalency [20].

## 2. Materials and Methods

### 2.1. Ethics Statement

All zebrafish experiments were performed in compliance with the Institutional Animal Care and Use Committee guidelines, National University of Singapore. HNoVs in stool samples were kindly provided by Singapore General Hospital. Type A saliva samples were collected from healthy adults. The human milk sample was donated by a blood type A woman during her breast-feeding period. The collection of both human saliva and human milk samples were under the ethical approval of National University of Singapore Institutional Review Board (NUS-IRB, reference code: N-20-035).

### 2.2. HNoV Virus-Like Particle (VLP) Binding Block Assay 

Type A saliva samples were boiled at 95 °C for 10 min, followed by centrifugation (Eppendorf, Hamburg, Germany) at 10,000× *g* for 5 min. The supernatants were diluted at 1:1500 by carbonate:bicarbonate buffer (pH 9.4) and used to coat 96-well microtiter plates (100 µL/well) at 4 °C overnight. After being washed once with phosphate-buffered saline (PBS) containing 0.05% Tween 20 (PBS-T), the plate was blocked with 5% nonfat dried milk (Blotto, 200 µL/well) at 37 °C for 1 h. NoV GII.4 VLPs (1.1 mg/mL, purity greater than 95%, The Native Antigen Company, OXFORD, UK) were pre-incubated with the fucoidan (from *Fucus vesiculosus*, Sigma-Aldrich, final concentrations at 2.40 and 24.00 mg/mL), 2′-FL (Sigma-Aldrich, St. Louis, MI, USA, final concentrations at 2.40 and 24.00 mg/mL) or human milk at room temperature for 1 h, diluted with 5% nonfat dried milk to a final VLP concentration of 1 µg/mL and added to the plates at 100 µL/well for incubation at 37 °C for 1 h. As the positive control, NoV GII.4 VLPs were pre-incubated with PBS and diluted with 5% nonfat dried milk to the same concentration (1 µg/mL) before adding to the plates. After being washed three times with PBS-T, the bound NoV capsid proteins were detected with mouse anti norovirus GII (1:1000; The Native Antigen Company) and by adding horseradish peroxidase conjugated with goat anti-mouse IgG (1:1500; Sigma Aldrich). Horseradish peroxidase activity was detected with a TMB (3,3′,5,5′-tetramethylbenzidine) kit (Sigma Aldrich), and the signal intensities (the optical density at 450 nm [OD450]) were read with a Multiskan Sky plate reader (Thermo Fisher Scientific, Waltham, MA, USA). OD_450_ of the blank controls were below 0.10.

### 2.3. Zebrafish Maintenance and Injection of Zebrafish Larvae with hNoV in the Yolk

Wild type adult zebrafish were maintained in the aquatic facility of National University of Singapore with water temperature at 28 °C and 14/10 h light/dark cycle. Fertilized eggs were collected from adults placed in mating cages and kept in petri dishes containing embryo medium E3 (5mM of NaCl, 0.17 mM KCl, 0.33 mM of CaCl_2_, 0.33 mM MgSO_4_).

The two NoV GII samples used in this study were provided by Singapore General Hospital and genotyped in our laboratory as GII.P16-GII.4 and GII.P31 (GII.Pe)-GII.17 over the B-C genome region as reported by Anderson et al. and Kojima et al. and with the use of Norovirus Automated Genotyping Tool developed by Kroneman et al. [25,26,27]. In order to prepare the virus suspension for injecting to zebrafish larvae, an aliquot of 100 mg of each stool sample was suspended in 1 mL of sterile PBS, thoroughly vortexed and centrifuged for 5 min at 9000× *g*; the supernatant was harvested and stored at −80 °C until further use.

The injection was performed according to Van Dycke et al., (2019) with slight modifications [14]. Briefly, 3 days post-fertilization (dpf) zebrafish larvae were anaesthetized for 2–3 min in embryo medium E3 containing 0.2 mg/mL Tricaine (Sigma-Aldrich). Thereafter, the anaesthetized zebrafish larvae were transferred to a petri dish with grooves of a mold imprint (6 rows of V-shaped grooves) in 1.5% agarose, aligned so that they were lying on their dorsal side with the yolk facing upwards. Three nL of the viral suspension (with or without pre-incubation with fucoidan or 2′-FL at room temperature for 1 h) was injected into the yolk of each zebrafish larva with a pulled borosilicate glass capillary needle. After the injection, the zebrafish larvae were transferred to a petri dish containing E3 and further maintained at 29.5 °C. At 1, 48 and 72 h post-injection (hpi), regarded as 0, 2 and 3 days post-injection (dpi), the zebrafish larvae were harvested with 10 larvae pooled as one sample, washed and collected in 2 mL tubes containing 0.5 mm yttria-stabilized zirconium oxide beads, Lysing Matrix Y (MP Biomedicals, USA).

After the injection, the survivability of the zebrafish larvae were monitored constantly until being harvested. A cut-off of 90% survival was applied guarantee the injection was performed properly and the matrix of the injected virus suspension did not manifest toxicity to bias the results.

### 2.4. Tissue Homogenization, RNA Extraction and RT-qPCR for Detection of hNoV

The zebrafish larvae collected in Lysing Matrix Y tubes were homogenized with 3 cycles of 15 s at 6500 RPM with rest intervals of 60 s by FastPrep^TM^ 24-5G tissue-cell homogenizer (MP Biomedicals, USA). The homogenates were clarified by centrifugation at 9000× *g* for 5 min and RNA was extracted using RNeasy Mini Kit (Qiagen, Hilden, Germany) following the manufacturer’s protocol.

RT-qPCR analyses of NoV GII were carried out using GoTaq^®^ Probe 1-Step RT-qPCR System (Promega, Madison, Wisconsin, United States). Primers and the FAM/TAMRA-labelled probe of NoV GII were according to ISO 15216-1:2017 [28]. Forward primer QNIF2: 5′-ATGTTCAGRTGGATGAGRTTCTCWGA-3′; reverse primer G2SKR: 5′-TCGACGCCATCTTCATTCACA-3′; probe QNIFs: 5′ FAM-AGCACGTGGGAGGGCGATCG-3′ TAMRA. Cycling conditions were 45 °C for 15 min, then 95 °C for 10 min, followed by 40 cycles with 95 °C for 15 s and 60 °C for 30 s in each cycle. Cycle threshold (Ct) values were determined during RT-qPCR analysis using StepOneTM system (Applied Biosystems, USA).

Double-stranded DNA (dsDNA) containing the specific primers-probe binding sites were synthesized for NoV GII and cloned into the pGEM-T Vector (Promega) resulting in the NoV-GII plasmids. The plasmids with inserts were purified by using a Plasmid Midi Kit (Qiagen). The plasmid concentration was determined by photospectroscopy at 260 nm using the BioDrop Duo^TM^ spectrophotometer (BioDrop, United Kingdom). Ten-fold serial dilutions ranging from 5 × 10^6^ to 5 copies of all positive control plasmids were used to prepare a standard curve and enumerate the NoV GII detected from the zebrafish larvae.

### 2.5. Statistical Analysis

Statistical analyses were performed using the SPSS 22 software for Windows (SPSS, Inc., Chicago, IL, USA). Student’s t-tests were performed on the VLPs binding block assay, and the nonparametric Mann-Whitney U tests were performed on the in vivo zebrafish model. Significant differences were considered when *p* was <0.05.

## 3. Results

### 3.1. Both Fucoidan and 2′-FL Were Able to Block the Binding of hNoV VLPs to HBGAs in Type A Saliva

Pre-incubation of hNoV GII.4 VLPs with both fucoidan and 2′-FL were able to decrease the binding affinity of hNoV GII.4 VLPs to type A saliva coated on plates in a dose-dependent manner (Figure 1). At both concentrations of 2.40 and 24.00 mg/mL, the reducing effects induced by 2′-FL were significantly (*p* < 0.05) more prominent than fucoidan. Last, a human milk sample collected from a type A secretor showed more efficient binding block effect than 24.00 mg/mL of fucoidan and 2′-FL (Figure 1).

### 3.2. HNoV GII.P16-GII.4 Replicated in Zebrafish Larvae More Consistently Than GII.P31-GII.17 at 2 dpi

In order to validate if the zebrafish larvae could indeed support the replication of hNoVs with our experimental set-up and hNoV samples, two NoV genogroup II clinical (stool) samples belonging to GII.P16-GII.4 (8-log genome copies/g stool) and GII.P31-GII.17 (9-log genome copies/g stool) were tested by injecting 3 nl of the virus suspension in the yolk of each larva at 3 dpf. At 0, 2 and 3 dpi, 10 injected larvae were pooled as one sample and proceeded to homogenization, RNA extraction and RT-qPCR for hNoV GII detection.

As a result, for hNoV GII.P16-GII.4, a low injection dose of ~2-log genome copies per larva (detected as <2-log genome copies per sample as shown in Table 1A) could lead to consistent virus replication to >4-log genome copies per sample at 2 dpi for all tested five samples in four independent experiments (Table 1A). At 3 dpi, two out of the five samples dropped with >2-log genome copies/sample and thus in the following studies, 2 dpi was selected as the time point to evaluate the anti-hNoV effect for hNoV GII.P16-GII.4.

As for GII.P31-GII.17, although it was able to start with higher injection doses due to the higher virus loads in the stool sample, replication of only ~1-log genome copies/sample was observed in two out of the five samples, while no viral replication was observed for the rest (Table 1B), showing that GII.P31-GII.17 is less prominent than GII.P16-GII.4 to be used to evaluate the anti-hNoV effect.

### 3.3. Fucoidan, but Not 2′-FL, Inhibited hNoV GII.P16-GII.4 Replication in Zebrafish Larvae

The effects of fucoidan and 2′-FL on hNoV replication in zebrafish larvae were investigated starting with 24.00 mg/mL, as pre-incubation with both carbohydrates at this concentration induced significant reductions (*p* < 0.05) of NoV GII.4 VLPs binding to type A saliva (Figure 1). However, 24.00 and 12.00 mg/mL fucoidan showed toxicity when injected to the zebrafish larvae and the larvae were only able to maintain the survival rates above 90% at 2 dpi when injected with low concentrations of fucoidan from 6.00 mg/mL downwards.

At 2 dpi, the zebrafish larvae groups injected with hNoV GII.P16-GII.4 suspension pre-incubated with 6.00 and 2.40 mg/mL fucoidan showed no virus replication as the RT-qPCR detection results for hNoV GII were below the detection limit (<1.3-log genome copies/sample), whereas lower concentration of fucoidan at 0.24 mg/mL as well as the positive control group still supported viral replication to >4-log genome copies/sample (Table 2A).

As for 2′-FL, when pre-incubated with hNoV GII.P16-GII.4 suspension and injected to the zebrafish larvae, no inhibition was observed even at the high concentration of 24.00 mg/mL (Table 2B).

Next, we selected the highest possible concentration at 6.00 mg/mL to test the effect fucoidan and 2′-FL in parallel. During all four independent experiments performed in four different weeks, the groups of 6.00 mg/mL fucoidan were three times detected with below detection limit (<1.3-log genome copies/sample) and once slightly above (1.8-log genome copies/sample), and the groups of 6.00 mg/mL 2′-FL did not show any significant inhibition in comparison with the positive controls (*p* > 0.05) (Figure 2).

### 3.4. Saliva But Not Human Milk from Type A Secretors Inhibited hNoV GII.P16-GII.4 Replication in Zebrafish Larvae

The saliva and human milk used in this study were collected from both type A secretors. The human milk was able to block the binding of NoV GII.4 VLPs to saliva even more efficiently than 24.00 mg/mL fucoidan and 2′-FL (Figure 1). When the hNoV GII.P16-GII.4 viral suspensions were pre-incubated with the saliva before injecting into the zebrafish larvae, no virus replication was observed (2.4-log genome copies/sample at 0 dpi and 2.2-log genome copies/sample at 2 dpi, Table 2C), whereas the group pre-incubated with human milk still supported viral replication to >4-log genome copies/sample (Table 2D).

## 4. Discussion

Abundant evidence from dedicated research has shown that recognition of HBGAs is essential for hNoV to cause infection [7,8,29]. As α-fucose has been determined to be an important glycan epitope of HBGAs binding to hNoVs [20], fucosylated carbohydrates including both fucoidan and 2′-FL have been reported to be able to inhibit the binding of hNoVs to HBGAs and thus proposed as potential anti-hNoV candidates [10]. Fucoidan is a fucose-containing sulphated polysaccharide commonly found from brown algae with widely reported bioactivities, such as antioxidant, anticancer, antiviral and antibacterial activities [30]. Recently, Kim et al. (2020) have found that oral administration of *Laminaria japonica* fucoidan to STAT1^−/−^ mice was able to improve their survival rates from murine norovirus infection and reduce virus loads in their feces [21]. 2′-FL is one of the most recognized human milk oligosaccharides [31] and has been applied in infant formula products on the markets [32]. The structural basis between 2′-FL and hNoVs have been clearly elucidated [22,23,24], which is believed to be the theoretical foundation to explain the protective effects of breast milk against hNoV infection. Therefore, both fucoidan and 2′-FL were tested in this study with the novel in vivo zebrafish larva platform in order to further investigate their anti-hNoV activities.

According to our results, although both fucoidan and 2′-FL were able to block hNoV GII.4 VLPs binding to type A saliva as expected, only fucoidan, but not 2′-FL, was able to inhibit the replication of hNoV GII.P16-GII.4 in zebrafish larvae. To the best of our knowledge, this is the first time showing that blocking of HBGA binding sites in vitro may not necessarily inhibit hNoV replication in vivo. As an explanation, we assume that fucoidan, with much higher molecular weight than 2′-FL, may supply a much more complete “shield” for the virions to prevent the interactions between the viruses with other downstream receptors/co-factors; whereas 2′-FL binding to the HBGA binding pockets did not manifest such an effect. Although HBGA is shown to have a close correlation with hNoV infection, the pathogenesis of hNoVs is still under investigation [33] and the exact role that HBGAs play in the hNoV infection remains largely unknown. Several proteins from human biological components have been reported with hNoV affinity, suggesting that there could be other essential receptors next to HBGA involved in hNoV infection [34]. Our result supports this point of view from a brand new angle. One may argue that the infection mechanism in zebrafish larvae may differ from infection in human, which is indeed pending to be studied in-depth in the future. However, as the virus used in this study is a genuine hNoV strain with clinical evidence from the supplier, there is a high chance that the zebrafish larvae share similarities in the routes supporting hNoV replication with human being.

In order to investigate further on the correlation between hNoV-HBGA binding and hNoV replication inhibition, we tested saliva and human milk, the two biological samples with known affinities to hNoVs. When pre-incubated with hNoV GII.P16-GII.4 and injected to zebrafish larvae, only the saliva, but not the human milk sample, tested in this study (collected from type A secretors) was able to inhibit the replication of hNoV. Saliva and human milk are complicated mixtures with not only oligosaccharides but also, more importantly, glycoproteins [35]. Although both contain HBGAs, the carbohydrate profiles of saliva and human milk have been indeed reported to be different [36]. In this study, it seems only the saliva but not the human milk sample was able to shield the hNoV virions from being recognized by the downstream receptor/co-factors causing infection in the zebrafish larvae, which remain to be revealed in the future. It must be noticed that this result may still vary depending on the phenotype of the biological sample donors. The data presented in this study is only to complement the results of fucoidan and 2′-FL, demonstrating the discrepancies between HBGA-NoV affinity and NoV replicating inhibition in zebrafish larvae.

In summary, with the use of a zebrafish larva platform, this study validated the anti-hNoV activity of fucoidan originated from brown algae *Fucus vesiculosus* and demonstrated the possible needs of higher molecular weights, other than oligosaccharides, such as 2′-FL for fucosylated carbohydrates, to exert the anti-hNoV effect. The zebrafish larva tool has shown excellent sensitivity in our hands as we have been using a virus stool sample (GII.P16-GII.4) with much lower virus load (~10^8^ genome copies/g stool) in comparison with the report by Van Dycke et al. (2019), of which the stool samples were with virus loads generally higher than 10^10^ genome copies/g stool [14]. This zebrafish larva platform still suffers from a few limitations, such as short-duration infection, which only lasts for a few days, and infeasibility of virus inoculation via immersion [14]. Thus, it is not possible to test more time points of fucoidan administration to mimic the genuine scenarios of human infection and protection. However, with the use of an in vivo model, especially by showing the controversial results between in vivo and in vitro tests of 2′-FL, this study serves as one step forward in exploring potential anti-hNoV candidates with more solid evidence.

## Figures and Tables

**Figure 1 viruses-13-00461-f001:**
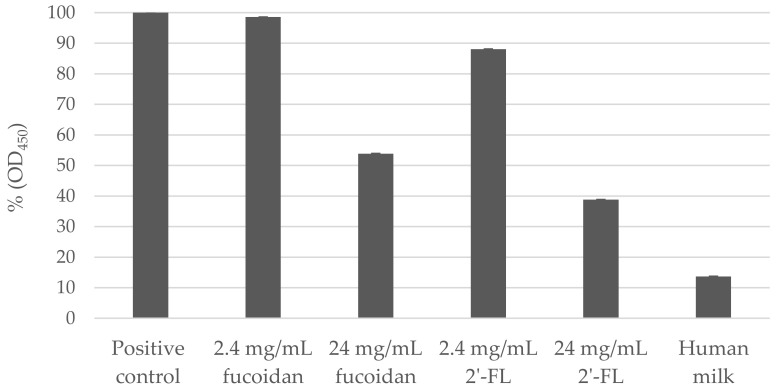
Percentage (%) of norovirus (NoV) GII.4 virus-like particles (VLPs) pre-incubated with fucoidan, 2′-Fucosyllactose (2′-FL) or human milk binding to type A saliva (OD_450_) in comparison to positive control (NoV GII.4 VLPs without carbohydrate blocking). Each column represents the average of triplicates, and each error bar indicates the standard deviations.

**Figure 2 viruses-13-00461-f002:**
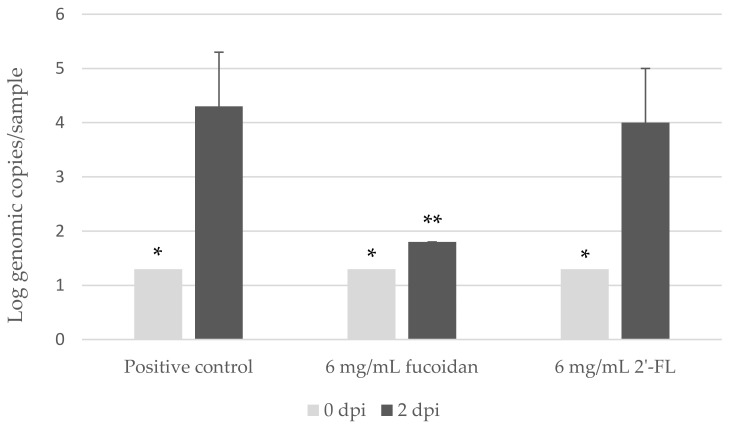
Detection of human norovirus (hNoV) GII.P16-GII.4 with and without pre-incubation with fucoidan or 2′-Fucosyllactose (2′-FL) from zebrafish larva at 0 and 2 dpi. Each sample contains a pool of 10 larvae. Each column represents the average of four independent experiments conducted in four different weeks, and each error bar indicates the standard deviations. * Below the limit of detection (1.3 log genome copies/sample). ** Two out of three samples were below the limit of detection.

**Table 1 viruses-13-00461-t001:** Detection of hNoV GII.P16-GII.4 (**A**) and GII.P31-GII.17 (**B**) in zebrafish larvae (log genome copies/sample) at different days post injection (dpi). Each sample contains a pool of 10 larvae. The limit of detection was 1.3 log genome copies/sample.

**(A)**				
**Batch**	**Sample**	**Log Genome Copies/Sample**		
		**0 dpi**	**2 dpi**	**3 dpi**
Batch 1	Sample 1	1.7	4.7	4.7
	Sample 2	1.5	4.4	4
Batch 2	Sample 1	1.9	4.6	4.8
Batch 3	Sample 1	<1.3	4.8	2.5
Batch 4	Sample 1	<1.3	4.9	2.6
**(B)**				
**Batch**	**Sample**	**Log Genome Copies/Sample**		
		**0 dpi**	**2 dpi**	**3 dpi**
Batch 1	Sample 1	2.5	n.a.	3.4
	Sample 2	2.5	n.a.	2.3
Batch 2	Sample 1	4.3	n.a.	4.4
Batch 3	Sample 1	2.8	2.9	n.a.
Batch 4	Sample 1	3	4.8	n.a.

**Table 2 viruses-13-00461-t002:** Detection of hNoV GII.P16-GII.4 pre-incubated with different concentrations of fucoidan (**A**), 2′-FL (**B**), saliva (**C**) and human milk (**D**) in zebrafish larvae (log genome copies/sample). Each sample represents a pool of 10 larvae. The limit of detection was 1.3 log genome copies/sample.

**(A)**			
**Concentration of Fucoidan**	**Zebrafish Larvae Survival Rate at 2 dpi**	**Log Genome Copies/Sample**	
		**0 dpi**	**2 dpi**
24.0 mg/mL	<90%	<1.3	n.a.
12.0 mg/mL	<90%	<1.3	n.a.
6.00 mg/mL	>90%	<1.3	<1.3
2.40 mg/mL	>90%	<1.3	<1.3
0.24 mg/mL	>90%	<1.3	4.3
0	>90%	<1.3	4.7
**(B)**			
**Concentration of 2′-FL**	**Zebrafish Larvae Survival Rate at 2 dpi**	**Log Genome Copies/Sample**	
		**0 dpi**	**2 dpi**
24.0 mg/mL	>90%	1.4	5.3
6.00 mg/mL	>90%	1.9	3.9
2.40 mg/mL	>90%	2.3	4.2
0	>90%	1.9	4.1
**(C)**			
	**Zebrafish Larvae Survival Rate at 2 dpi**	**Log Genome Copies/Sample**	
		**0 dpi**	**2 dpi**
Saliva (type A secretor)	>90%	2.4	2.2
Control	>90%	1.9	4.1
**(D)**			
	**Zebrafish Larvae Survival Rate at 2 dpi**	**Log Genome Copies/Sample**	
		**0 dpi**	**2 dpi**
Human milk (type A secretor)	>90%	<1.3	5.7
Control	>90%	<1.3	5.4

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
