# Peer review of "Fucoidan But Not 2′-Fucosyllactose Inhibits Human Norovirus Replication in Zebrafish Larvae"

_viruses, 2021, doi:10.3390/v13030461_

Round 1
Reviewer 1 Report
The topic of the paper “Fucoidan but not 2’-Fucosyllactose could inhibit human no-rovirus replication in zebrafish larvae” is interesting.
The authors indicate that human noroviruses are increasingly identified as an important cause of acute gastroenteritis outbreaks associated with great disease burden worldwide.
This study serves as one step forward in exploring potential anti- noroviruses candidates with more solid evidence.
The manuscript is well written and the text is easy to read.
The results are consistent and clearly presented.
So, I recommend the publication of the present manuscript, but I have some small observations:
- The reference numbers 1, 8, 12, 15, 16, 17, 19, 21, 23, 24, 25, 29-32, 34 are written different from the others!!!!
Please to correct these aspects!
Author Response
Answer: Thanks for reviewing our work and confirming our efforts. We have went through the whole manuscript for further editing and the references have been corrected accordingly.
Reviewer 2 Report
Comments for “Fucoidan but not 2'-Fucosyllactose could inhibit human norovirus replication in zebrafish larvae”
Authors: Malcolm Turk Hsem Tan, Yan Li, Mohamad Eshaghi Gorji, Zhiyuan Gong, and Dan Li
Introductory statement
The paper titled “Fucoidan but not 2'-Fucosyllactose could inhibit human norovirus replication in zebrafish larvae” by Dr. Turk Hsern Tan et al. describes a novel in vivo approach to testing antiviral compounds against the human norovirus. The study addresses an important issue, since safe, effective antiviral compounds are lacking for most viruses, including noroviruses. One major roadblock to assessing antiviral compounds is a lack of biologically relevant and tractable models. Here, the researchers take a very clear and straightforward approach to studying norovirus infection by using in vitro binding assays and in vivo viral replication assays in a zebrafish larva model. This paper promises to be an important step in broadening the ability to assess norovirus biology, since it is one of the very few models where human norovirus particles actively replicate within cells.
Comments and Suggestions for the Authors
Major Comments
- The manuscript would benefit from some copyediting, since some of the language is awkward. Although the text is certainly readable, I would recommend a grammatical editing process.
- Please list the concentration of fucoidan and 2'-FL used in the VLP binding block assays in the Methods section. They are listed in the figures, but it would be helpful to have that detail in the Methods.
- Please list more detail about the positive control for the blocking assay (was it PBS? Water?)
- Please add a bit more detail about where the stool samples came from and who typed them.
- The toxicity findings for the carbohydrates in the zebrafish larvae before the norovirus experiments is compelling and important. These pre-experiments are not shown. I would recommend adding a table or figure with these results, or perhaps in a supplemental file. It’s not entirely necessary, but I think it would be helpful for researchers who may use this model in the future.
- For the experiments displayed in Figure 2 (incubating virus with carbohydrates before zebrafish inoculation), was a statistical test performed? The methods note that statistical tests were done for the zebrafish experiments, but they are not shown in the figures or explained in the text.
- There are no callouts in the text to Table 2C and Table 2D. Please add.
- Note that the information about 2'-FL and fucoidan that is contained in the first paragraph of the Discussion seems to be more pertinent for the Introduction. It's not imperative, but I might add a little more detail in the Intro about the significance of testing carbohydrates. Right now, the Intro focuses on the zebrafish model, which is important. But the question about carbohydrates as antiviral compounds is equally important.
Minor Comments
- As mentioned in the major comments, there are some formatting/punctuation errors that need to be fixed throughout, specifically, fixing the abbreviation for 2'-Fucosyllactose to 2'-FL instead of 2-'FL
- In the figure axes and in tables, the proper labeling is “log genome copies / sample”
- The title is a little awkward. The term “could” is accurate, but not typical for a title. If the title states the conclusions, I would make it an active statement (eg, inhibits, rather than could inhibit). But I think that a title that does not state the conclusions might be more relevant, depending on the journal’s guidelines and the editor’s preference. (eg, The Effect of Fucoidan and 2'-Fucosyllactose on Replication of Human Norovirus in a Zebrafish Larvae Model)
Author Response
Introductory statement
The paper titled “Fucoidan but not 2'-Fucosyllactose could inhibit human norovirus replication in zebrafish larvae” by Dr. Turk Hsern Tan et al. describes a novel in vivo approach to testing antiviral compounds against the human norovirus. The study addresses an important issue, since safe, effective antiviral compounds are lacking for most viruses, including noroviruses. One major roadblock to assessing antiviral compounds is a lack of biologically relevant and tractable models. Here, the researchers take a very clear and straightforward approach to studying norovirus infection by using in vitro binding assays and in vivo viral replication assays in a zebrafish larva model. This paper promises to be an important step in broadening the ability to assess norovirus biology, since it is one of the very few models where human norovirus particles actively replicate within cells.
Answer: Thanks for reviewing our work and confirming our efforts.
Comments and Suggestions for the Authors
Major Comments
The manuscript would benefit from some copyediting, since some of the language is awkward. Although the text is certainly readable, I would recommend a grammatical editing process.
Answer: We have went through the whole manuscript and made further editing accordingly.
Please list the concentration of fucoidan and 2'-FL used in the VLP binding block assays in the Methods section. They are listed in the figures, but it would be helpful to have that detail in the Methods.
Answer: Done.
Please list more detail about the positive control for the blocking assay (was it PBS? Water?)
Answer: Done. “As the positive control, NoV GII.4 VLPs were pre-incubated with PBS and diluted with 5% nonfat dried milk to the same concentration (1 µg/ml) before adding to the plates.”
Please add a bit more detail about where the stool samples came from and who typed them.
Answer: Done. “The two NoV GII samples used in this study were provided by Singapore General Hospital and genotyped in our laboratory”.
The toxicity findings for the carbohydrates in the zebrafish larvae before the norovirus experiments is compelling and important. These pre-experiments are not shown. I would recommend adding a table or figure with these results, or perhaps in a supplemental file. It’s not entirely necessary, but I think it would be helpful for researchers who may use this model in the future.
Answer: We believe this is very good point. Indeed this survival rate is important as a premise to conduct the study. In fact, it is also to control the injection was conducted successfully as it needs some training and skills. The stool matrix itself may also be toxic to the fish (not observed in this study though). Thus we have added the following text in materials and methods to be clear with this point. “After the injection, the survivability of the zebrafish larvae were monitored constantly until being harvested. A cut-off of 90% survival was applied guarantee the injection was performed properly and the matrix of the injected virus suspension did not manifest toxicity to bias the results.”
For the experiments displayed in Figure 2 (incubating virus with carbohydrates before zebrafish inoculation), was a statistical test performed? The methods note that statistical tests were done for the zebrafish experiments, but they are not shown in the figures or explained in the text.
Answer: As shown in the text “the groups of 6.00 mg/ml 2-’FL did not show any significant inhibition in comparison with the positive controls (P>0.05)”. The other groups involve detection results below detection limits, thus it was not possible to perform statistical analysis.
There are no callouts in the text to Table 2C and Table 2D. Please add.
Answer: We are sorry for missing it in the text, it has been added accordingly.
Note that the information about 2'-FL and fucoidan that is contained in the first paragraph of the Discussion seems to be more pertinent for the Introduction. It's not imperative, but I might add a little more detail in the Intro about the significance of testing carbohydrates. Right now, the Intro focuses on the zebrafish model, which is important. But the question about carbohydrates as antiviral compounds is equally important.
Answer: Thanks for this comment but we feel it’s better to keep the current brief introduction over the carbohydrates and to elaborate more in detail in the discussion to facilitate for the readers comparing the previous reports with our results directly.
“Both carbohydrates have been reported as putative anti-hNoV candidates due to their fucose contents and thus specific hNoV binding ability [20, 21, 22, 23, 24]. Specifically, 2-’FL was observed to bind to NoV GII.4 and GII.10 capsid at the same pockets with HBGA [23]. Fucoidan was proposed as a more efficient anti-NoV candidate due to its fucose multivalency [20].”
Minor Comments
As mentioned in the major comments, there are some formatting/punctuation errors that need to be fixed throughout, specifically, fixing the abbreviation for 2'-Fucosyllactose to 2'-FL instead of 2-'FL
Answer: Done.
In the figure axes and in tables, the proper labeling is “log genome copies / sample”
Answer: All corrected.
The title is a little awkward. The term “could” is accurate, but not typical for a title. If the title states the conclusions, I would make it an active statement (eg, inhibits, rather than could inhibit). But I think that a title that does not state the conclusions might be more relevant, depending on the journal’s guidelines and the editor’s preference. (eg, The Effect of Fucoidan and 2'-Fucosyllactose on Replication of Human Norovirus in a Zebrafish Larvae Model)
Answer: Thanks for pointing out, however this statement is something we really want to emphasis and communicate with the readers, thus if the editor approves we would prefer to keep it. We adopted the suggestions and changed “could inhibit” to “inhibits”.